# Effects of day-to-day variation of *Opisthorchis viverrini* antigen in urine on the accuracy of diagnosing opisthorchiasis in Northeast Thailand

Chanika Worasith[1,2], Phattharaphon Wongphutorn[2,3], Chutima Homwong[2], Kulthida Y. Kopolrat[4], Anchalee Techasen[2,5], Raynoo Thanan[6], Chatanun Eamudomkarn[1,2], Chompunoot Wangboon[7], Narong Khuntikeo[2,8], Watcharin Loilome[2,6], Jiraporn Sithithaworn[5], Thomas Crellen[9,10], Paiboon Sithithaworn[1,2]*

1 Department of Parasitology, Faculty of Medicine, Khon Kaen University, Khon Kaen, Thailand, 2 Cholangiocarcinoma Research Institute, Khon Kaen University, Khon Kaen, Thailand, 3 Biomedical Science Program, Graduate School, Khon Kaen University, Khon Kaen, Thailand, 4 Faculty of Public Health, Kasetsart University Chalermphrakiat Sakon Nakhon Province Campus, Sakon Nakhon, Thailand, 5 Faculty of Associated Medical Sciences, Khon Kaen University, Khon Kaen, Thailand, 6 Department of Biochemistry, Faculty of Medicine, Khon Kaen University, Khon Kaen, Thailand, 7 School of Preclinical Science, Institute of Science, Suranaree University of Technology, Nakhon Rachasima, Thailand, 8 Department of Surgery, Faculty of Medicine, Khon Kaen University, Khon Kaen, Thailand, 9 Institute for Biodiversity, Animal Health and Comparative Medicine, University of Glasgow, Glasgow, United Kingdom, 10 Big Data Institute, Nuffield Department of Medicine, University of Oxford, Oxford, United Kingdom

* paib_sit@kku.ac.th

**Data Availability Statement:** The data are all contained within the paper in supporting files.

## Abstract

Antigen detection in urine using an enzyme-linked immunosorbent assay (ELISA) is more sensitive than fecal examination for diagnosis of opisthorchiasis and for assessment of the effects of drug treatment. It is not known whether day-to-day variation of urine composition, including levels of *Opisthorchis viverrini* antigen, influences the urine assay. We investigated this topic with the cooperation of participants from two localities in Northeast Thailand. Project participants were screened for parasite infections for three consecutive days using the quantitative formalin-ethyl acetate concentration technique (FECT) to detect *O. viverrini* eggs and the urine ELISA for detection of *O. viverrini* antigen. A subset of participants ($n =$ 801) with matched fecal and urine samples were analyzed for comparison of inter-day prevalence estimates and the performance of the urine assay compared against FECT for diagnosis of opisthorchiasis. The daily prevalence measured by the urine assay ranged between 29.0%-30.2% while those by FECT ranged between 11.9%-20.2%. The cumulative three-day prevalence estimate determined by the urine antigen assay was 30.3%, which was significantly higher than that by FECT (20.2%, $p < 0.05$). A significant positive correlation was found between the concentration of antigen in urine and fecal egg counts ($p < 0.001$). Overall, the urine assay had better diagnostic performance for opisthorchiasis than fecal examination by FECT. The high sensitivity plus negligible daily variation of *O. viverrini* antigen in urine indicates the utility of the urine assay for diagnosis, as well as population screening, of opisthorchiasis.

**Funding:** This work described here was supported by the Cholangiocarcinoma Research Institute, Khon Kaen University, and Fluke-Free Thailand, National Research Council of Thailand. This work was also supported by Thailand Center of Excellence for Life Sciences (TCELS). PS is supported by National Science, Research and Innovation Fund (NSRF) under the Basic Research Fund of Khon Kaen University through Cholangiocarcinoma Research Institute (CARI-BRF64-50). TC is supported by a Sir Henry Wellcome Postdoctoral Fellowship. This research was funded in whole, or in part, by the Wellcome Trust (Grant number 215919/Z/19/Z). The funders had no role in study design, data collection and analysis, decision to publish, or preparation of the manuscript.

**Competing interests:** The authors have declared that no competing interests exist.

## Introduction

The liver fluke *Opisthorchis viverrini* is the causative agent of opisthorchiasis; a major public health problem in the Mekong River Basin, especially in Thailand, the Lao People's Democratic Republic, Cambodia, and Vietnam [1, 2]. Chronic opisthorchiasis can induce hepatobiliary disease and cholangiocarcinoma (CCA) [3, 4]. A comprehensive control program aiming for elimination of the liver fluke is an important strategy for the reduction of CCA [5, 6].

Under the current transmission landscape of opisthorchiasis, an effective control program requires sensitive and specific diagnostic tools. Coprological methods are known to have high specificity but have many inherent drawbacks including requirement of experienced microscopists to distinguish *O. viverrini* eggs from those of small intestinal flukes and other trematodes [7]. Coprological methods are far less sensitive and reliable in light infections [8, 9]. For instance, no fecal eggs were detected by FECT in around two thirds of individuals harboring <20 worms, as shown by examination of the livers of autopsy subjects [10]. Multiple stool examinations (i.e. for three consecutive days) have been recommended to improve diagnostic sensitivity [11], but this is logistically impractical and costly.

Improvement in the diagnosis of opisthorchiasis by an immunological method (i.e. antibody detection) had been reported, but this does not permit past and current infection to be differentiated [12]. An alternative approach is to detect worm antigen; the presence of which implies current infection. Pioneering copro-antigen methods showed promising diagnostic potential in studies conducted during the 1990s [13, 14]. Almost a decade later, we have used a similar approach to demonstrate improved performance of copro-antigen detection in opisthorchiasis, particularly in light infections i.e., <50 egg/gm feces [15]. Subsequently, a new approach for antigen detection in urine was established and the assay offered a higher diagnostic performance than fecal examination [16, 17]. An initial community-based application of this method demonstrated correlations between the urine-antigen assay and fecal examination for estimates of prevalence and intensity of infection [16, 17]. In addition to screening for opisthorchiasis, the urine ELISA has been tested for monitoring outcomes of drug treatment and occurrence of reinfection in opisthorchiasis [18]. However, those studies relied on a single urine sample for analysis and interpretation of results. The influence of day-to-day variation on detectability and concentration of antigen in urine have therefore not been investigated.

In this study, we evaluated the day-to-day variation of *O. viverrini* antigen in urine over three consecutive days and the impact of this on diagnostic performance in comparison with fecal examination using FECT. We demonstrated no significant daily variation of the prevalence of opisthorchiasis, as measured by *O. viverrini* antigen in urine for three consecutive days. The persistent levels of antigen in urine provide further evidence to support the reliability of the urine assay for diagnosis of, and population screening for, opisthorchiasis.

## Materials and methods

### Study design, study areas and participants

This was a prospective study undertaken by following a cohort of participants for three consecutive days to evaluate the fluctuations of *O. viverrini* antigen in urine and of fecal egg output for quantitative and qualitative diagnosis of opisthorchiasis. The study began in December 2018 and ended in March 2019.

The selected study areas were known endemic areas for opisthorchiasis and cholangiocarcinoma in suburban communities in Northeast Thailand. The study areas were in Muang District, Khon Kaen Province (KKN) and Nong Kung Sri District, Kalasin Province (KSN).

Residents >15 years of age were invited to participate and recruitment involved written informed consent for collection of stool and urine samples from each participant.

## Clinical samples

Fecal samples (approx. 5 gm) and mid-stream first morning urine samples (approx. 10 ml) were obtained from the participants and were transferred to the laboratory within one day of collection. At the laboratory, fecal samples were processed for parasite examination using the formalin-ethyl acetate concentration technique (FECT). Urine samples were centrifuged at 1,500 rpm at 4˚C for 15 minutes and the supernatants were aliquoted and stored at -20˚C until used for urine assay.

## Fecal examination method

The procedure for quantitative FECT was as previously reported [19]. Three drops from the final fecal suspension were examined for the presence of *O. viverrini* and other parasites. To estimate intensity of infection, the egg counts were summed over the three drops, then multiplied by the total number of drops and divided by the mass of stool in grams multiplied by the number of drops examined to give eggs per gram of stool (EPG). The laboratory staff who undertook the procedure were blinded for sample IDs and laboratory data of the participants.

## Procedure for antigen detection in urine

The protocol for the *O. viverrini* antigen assay in urine using monoclonal antibody-based ELISA was similar to that previously described [16, 17]. Briefly, flat-bottomed 96-well microtiter plates (Maxisorb, NUNC, Denmark) were coated with 5 μg/ml of monoclonal antibody specific to antigens of *O. viverrini* and the plates were incubated overnight at 4˚C. On the next day, they were washed three times with a buffer containing 0.05% Tween 20 in PBS pH 7.4 (PBST) and uncoated sites were blocked with 5% dried skimmed milk in PBST. The plates were then incubated at 37˚C for 1 hour. Washing was repeated thrice with PBST, and samples of undiluted urine pre-treated with TCA (100 μl/well) were added to wells in duplicate and incubated at 37˚C for 2 hours. The plates were washed 5 times with PBST, and IgG rabbit anti-crude *O. viverrini* antigen was added and incubated at 37˚C for 1 hour. After plate washing, biotinylated goat anti-rabbit IgG conjugate (Invitrogen, USA) (1:4000 dilution, 100 μl/well) was added to each well and incubated at 37˚C for 1 hour. Thereafter, the plates were washed three times and streptavidin horseradish peroxidase (HRP)-conjugate (GE Healthcare, Buckinghamshire, UK) (1:5000 dilution, 100 μl/well) was added. After incubation and washing, a substrate solution, (*o*-phenylenediamine hydrochloride) solution (Sigma, St. Louis, MO, USA) (100 μl/well) was added and plates incubated for 20 min in the dark at room temperature. The reaction was stopped by the addition of 2M sulfuric acid (100 μl/well) and the plates were read using a plate reader (Tecan Sunrise Absorbance Reader, Austria) at 492 nm.

## Ethics statement

The human experimental protocol was approved by the Khon Kaen University Ethics Committee (reference number HE561478). Written informed consent was obtained from each project participant. Those positive for *O. viverrini* by FECT and/or the urine antigen assay were treated with a single oral dose of praziquantel (40 mg/kg body weight). In the case of other parasitic infections, participants were treated with appropriate drugs.

## Statistical analysis

Data obtained from the analyses of clinical samples were recorded and entered into the Microsoft Excel program and exported to SPSS statistical package SPSS v.28 (International Business Machines, USA) for all statistical analysis. The statistical distributions of the data were analyzed using the Kolmogorov Smirnov (K-S) test before further statistical testing. In the current study, the index test was urine ELISA, which was evaluated against a reference standard by FECT. Helminth species-specific fecal egg counts were transformed into number of eggs per gram of feces (EPG) to estimate intensity of infection. The ELISA optical density (OD) values were transformed to a ratio between the OD of the sample and the OD of reference positive urine samples, and antigen concentrations were calculated as ng/ml of urine based on standard curves [16, 17]. A urine sample was considered positive at an *O. viverrini*-antigen level of >20 ng/ml. Comparisons between daily or cumulative prevalence estimates determined by fecal examination and by urine ELISA were validated using McNemar's chi-square test. Chi-square tests were used to compare the prevalence of opisthorchiasis between age groups of the participants and between sexes. The correlation between fecal egg counts and *O. viverrini* antigen concentration in urine was assessed by a linear regression model. Two diagnostic reference standards were used for evaluation of urine ELISA and FECT. These were the primary standard method i.e., fecal examination (FECT) and the composite reference standard method (any participant positive by fecal examination and/or by urine ELISA was regarded as infected). Performances of diagnostic methods were evaluated as sensitivity and negative predictive values against the two reference standards. Agreement between the diagnostic methods was assessed using Cohen's kappa test. Interpretations of kappa values (κ) were graded as almost perfect agreement (κ = 0.81–1.00), substantial (κ = 0.61–0.80), moderate (κ = 0.41–0.60), fair (κ = 0.21–0.40), slight (κ = 0.00–0.20), and poor agreement (κ < 0.00) [20]. The results of statistical tests were considered significant when the P value was < 0.05. All graphs were constructed using GraphPad Prism version 8.0 (Graph Pad Software Inc, San Diego, CA, USA).

## Results

### Baseline parasitic infections

The flow diagram for recruitment of study participants (age >15 years) and analyses of clinical samples is shown in Fig 1. In total, 1,471 individuals were recruited and fecal examination by FECT was performed to determine prevalence of infection. Infection with *O. viverrini* was the most common parasitic disease detected at both localities, as shown in S1 Table. Apart from *O. viverrini*, other parasites found in this study included *Strongyloides stercoralis*, minute intestinal flukes, hookworm, *Taenia* sp., *Echinostoma* sp., *Hymenolepis diminuta*, *Enterobius vermicularis*, *Giardia lamblia* and *Blastocystis hominis*.

From the original cohort of participants, sufficient volumes of urine and fecal specimens for three consecutive days were obtained from 801 individuals (Fig 1). These consisted of 360 males and 441 females. Ages ranged from 21 to 86 years old (mean ± SD, 53.3 ± 10.9). There were no significant differences for sex (p > 0.05) and age (p > 0.05) between included (801 individuals) and excluded (670 individuals) subject groups.

The age-prevalence profiles of *O. viverrini* in KKN significantly increased with age when determined by FECT (Chi-square = 13.461, p<0.05) and urine assay (Chi-square = 8.639, p<0.05) (S2 Table). The prevalence of *O. viverrini* by urine assay was comparable to FECT in all age groups and between sex in KKN (p > 0.05). In KSN, the prevalence significantly increased with age in KKN as measured by FECT (Chi-square = 9.566, p < 0.05) and by urine

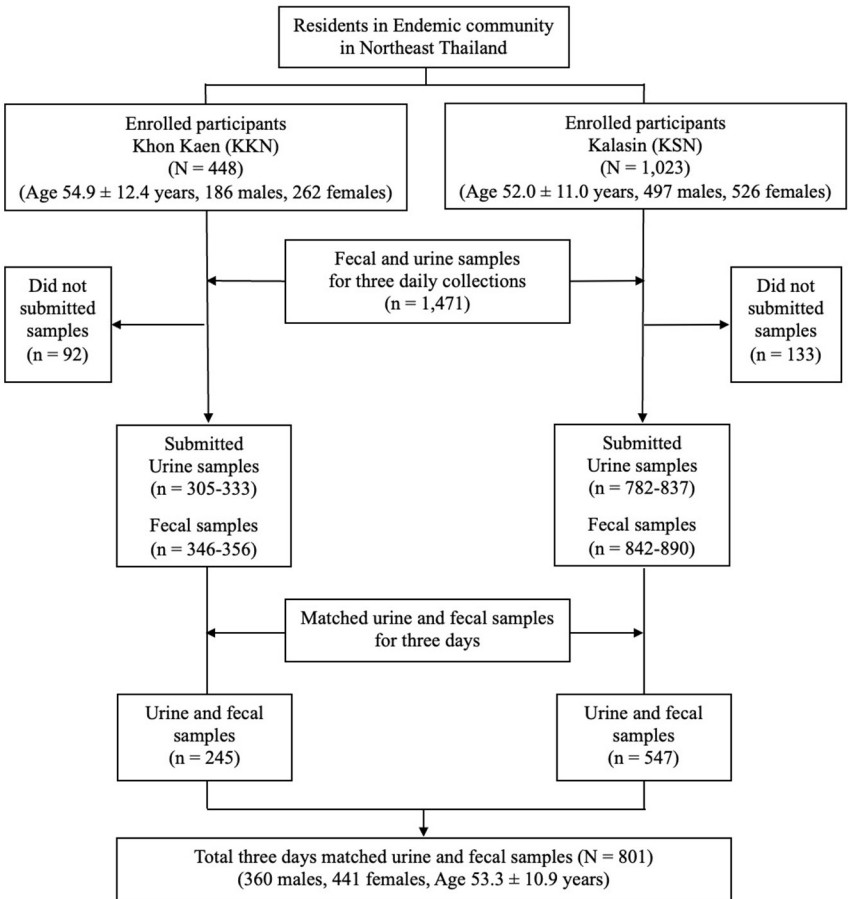

**Fig 1. Flow diagram for recruitment of study participants to examine the daily fluctuation of fecal egg counts and antigen levels in urine in opisthorchiasis.** The participants were residents from Muang District, Khon Kaen Province (KKN) and Nong Kung Sri District, Kalasin Province (KSN), Northeast Thailand.

antigen assay (Chi-square = 9.791, p < 0.05). A significantly higher prevalence was observed by urine assay compared with FECT in all age groups in KSN (Chi-square = 95.91, p < 0.001). There was a higher prevalence by urine assay than FECT in both males and females (chi-square test, p < 0.001) (S2 Table).

### Daily prevalence and intensity of *O. viverrini* infection by FECT and urine assay

The daily prevalence of *O. viverrini* by FECT varied between 13.9% and 17.7% for KKN and from 7.9% to 9.1% for KSN. The daily prevalence, as assessed by urine ELISA, varied between 26.0% and 27.3% for KKN and from 28.3% to 29.0% for KKN (Fig 2). As shown in Table 3, the cumulative 3-day prevalence according to FECT significantly increased as a result of sampling on days 2 and 3 in both KKN and KSN and the two sites combined (p < 0.001). The prevalence of *O. viverrini* infection by urine antigen assay was comparable over the three consecutive days in each locality separately or combined (Fig 2).

Based on fecal egg counts and urine antigen concentrations of *O. viverrini*, variations occurred between days in both KKN and KSN over the 3-day period. Significant differences in EPG (p < 0.001) between days of examination were found in both KKN and KSN. For urine

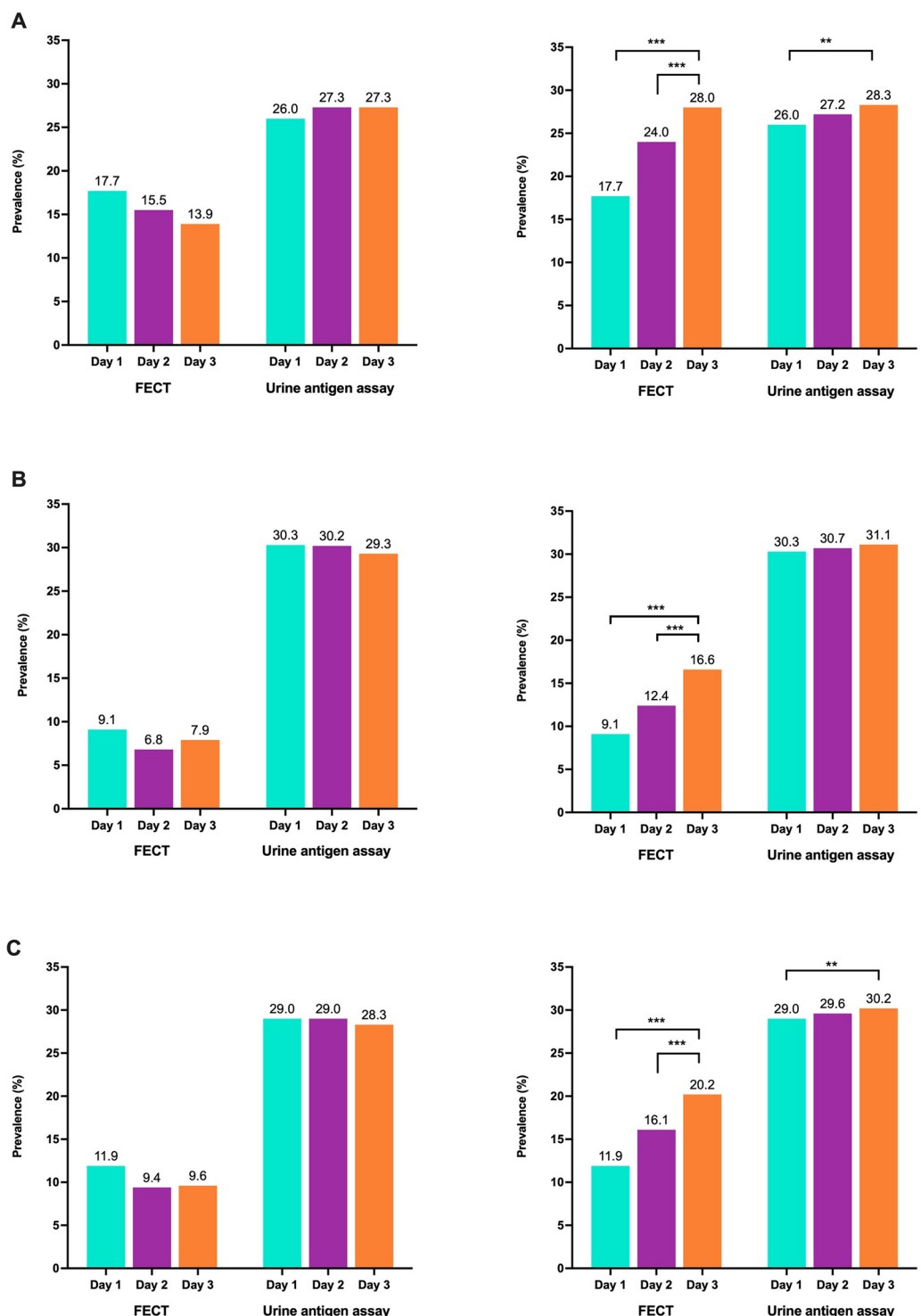

**Fig 2. Prevalence of *O. viverrini* infection determined by FECT and urine assay for three consecutive days.** The daily prevalence of opisthorchiasis (left panel) and the cumulative prevalence (right panel) are illustrated according to locality. (A) KKN; Muang District, Khon Kaen Province. (B) KSN; Nong Kung Sri District, Kalasin Province. (C) Both sites combined. McNemar Test, ** P < 0.01, *** P < 0.001.

**Table 1. Daily variations in intensity of infection measured by fecal egg counts and antigen concentration in urine in Muang District, Khon Kaen Province (KKN), Nong Kung Sri District, Kalasin Province (KSN) in Northeast Thailand.** Data shown are geometric means of EPG and antigen concentrations (ng/ml) in urine.

| Days of examination | Fecal egg count (EPG) GM ± SE (Range) | | |
| --- | --- | --- | --- |
| | KKN (n = 245) | KSN (n = 547) | Total (n = 801) |
| Day 1 | 5.3 ± 1.9 (2–52) | 8.2 ± 2.6 (2–904) | 7.1 ± 2.5 (2–904) |
| Day 2 | 8.9 ± 3.1[†] (2–98) | 10.2 ± 2.8 (2–173) | 9.7 ± 2.9 (2–173) |
| Day 3 | 9.9 ± 2.5[†, ‡] (2–65) | 18.9 ± 3.1[†, ‡] (4–669) | 14.2 ± 2.9[†, ‡] (2–669) |
| Kruskal-Wallis test | < 0.001 | < 0.001 | < 0.001 |
| | Antigen concentration in urine (ng/ml) GM ± SE (Range) | | |
| Day 1 | 24.9 ± 1.3 (19.4–74.7) | 28.3 ± 1.5 (19.4–146.8) | 27.5 ± 1.5 (19.4–146.8) |
| Day 2 | 27.3 ± 1.4 (19.5–80.6) | 25.1 ± 1.3[†] (19.4–99.6) | 25.7 ± 1.3 (19.4–99.6) |
| Day 3 | 26.6 ± 1.3 (19.4–87.9) | 25.2 ± 1.3[†] (19.5–92.2) | 25.6 ± 1.3 (19.4–92.2) |
| Kruskal-Wallis test | > 0.05 | < 0.001 | > 0.05 |

GM ± SE, Geometric mean ± standard error,

[†] = day 2 and day 3 compare with day 1 at P < 0.05;

[‡] = day 1 and day 2 compared with day 3 at P < 0.05 by Mann-Whitney U test.

Kruskal-Wallis test of transformed daily data (P > 0.05).

antigen concentrations, a significant variation between days was observed only in KSN (Table 1). When data for the two localities were combined, fecal egg counts showed significant daily variation but urine antigen levels did not.

## Correlations between of *O. viverrini* fecal egg counts and antigen concentrations in urine

For quantitative analysis, there were significant positive correlations between EPG and urine antigen concentrations as shown in Fig 3 (Spearman's range correlation, p < 0.001). When separated by locality or combined locality, similar correlations were observed (S3 Table).

As shown in S4 Table, to assess the effect of fecal egg counts (i.e. intensity of infection) on antigen detection in urine, the participants were separated into 4 different intensity groups based on fecal egg counts by FECT. In egg-negative (EPG = 0) individuals, 3.8%, 20.2% and 15.3% were positive according to urine ELISA in KKN, KSN and combined sites, respectively. In egg-positive groups (EPG>0), 89.9%-100% were positive according to urine ELISA in KKN, 84.1%-100% in KSN and 86.4%-100% in combined sites.

## Diagnostic performance of fecal examination and urine assay

Table 2 shows performance statistics for FECT and urine assay. These include sensitivity and negative predictive values (NPV) using FECT as a primary reference standard and combined FECT and urine assay results as a composite reference standard. In KKN, both the sensitivity and NPV of FECT and the urine assay were similar (>90%) using either the primary (diagnosis by the other test for three consecutive days as a gold standard) or composite reference standard. In KSN, the performance of FECT using either the primary or composite reference standard was 46.4%-50% sensitivity compared with 86.8%-93.4% sensitivity for the urine assay. For both sites combined, the diagnostic performance of FECT was 59.5% and 62.3% sensitivity using primary and composite standards, respectively, with high NPV (>80%). For the urine assay, sensitivity was 88.8% compared with the primary standard and 93.1% compared with the composite standard.

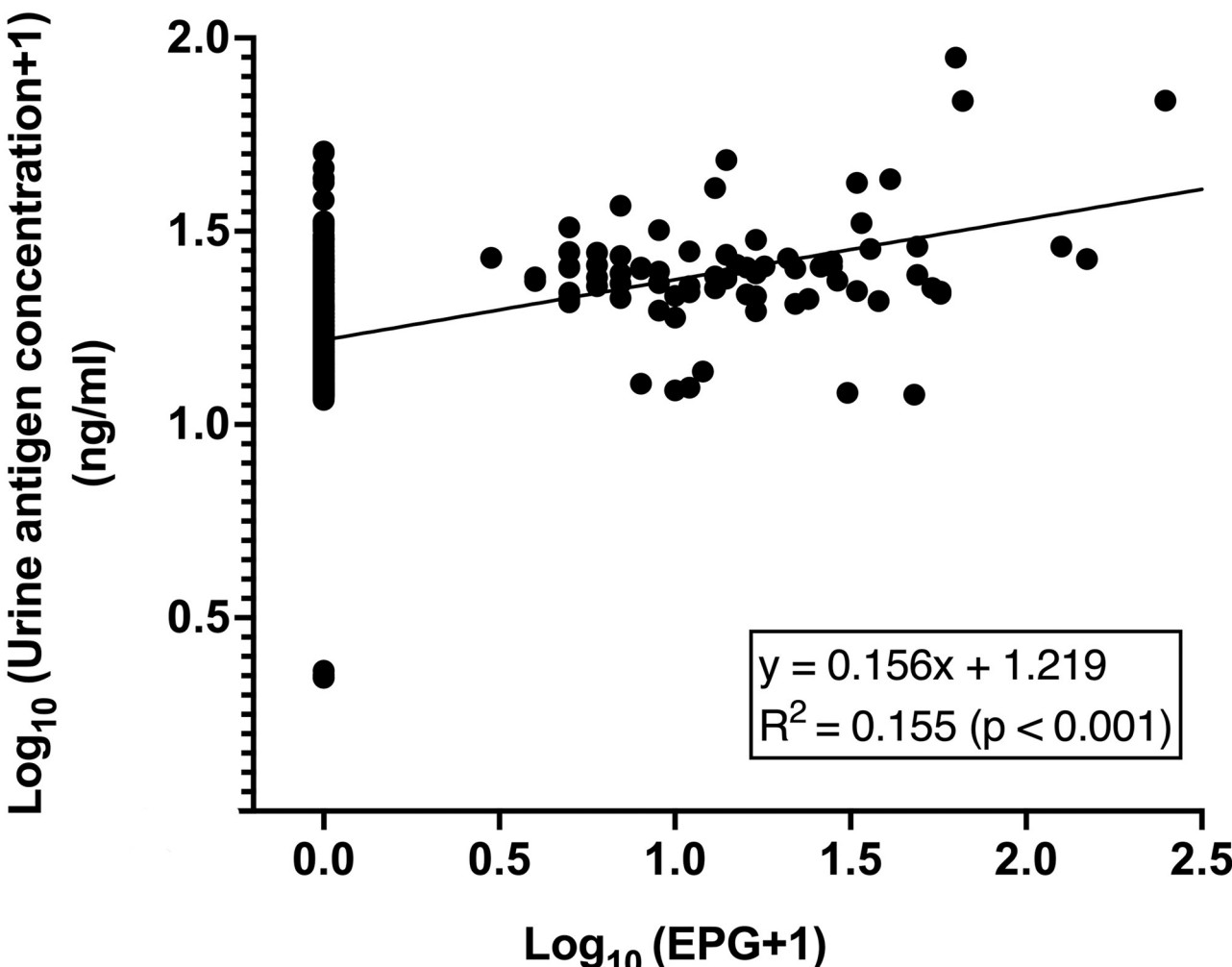

**Fig 3. Relationship between log-transformed fecal egg counts of *O. viverrini* (EPG) and log-transformed concentrations of antigen in urine (ng/ ml) in 801 individuals from Northeast Thailand.** Data points shown are observed values, where the egg counts plus one are $\log_{10}$ transformed (x-axis) and the antigen concentrations are $\log_{10}$ transformed (y-axis). The solid line represents a best-fit linear-regression with an intercept of 1.219 and a slope of 0.156 (Spearman's range correlation test, $p < 0.001$).

As shown in Table 3, for FECT the agreement in diagnosis of *O. viverrini* infection for daily prevalence estimates was fair (kappa value 0.350–0.422). For cumulative (three-day estimates), there were good to excellent agreements (kappa = 0.693–0.824). In the urine assay, there was an excellent agreement for daily and cumulative three-day estimates (kappa > 0.900). Between FECT and urine antigen assay, the agreements were fair to moderate in terms of daily prevalence and were moderate to good for cumulative three-day prevalence estimates.

## Discussion

In this study we compared the inter-daily variation in diagnosis of *O. viverrini* by either the standard FECT diagnostic or the urine antigen test. The urine antigen assay was consistent in results generated across the three days, while the fecal examination by FECT exhibited significant variation. Significant inter-day differences were noted in prevalence estimates as well as average fecal egg counts (EPG). No significant inter-day differences were observed for the

**Table 2. Performance statistics for fecal examination by FECT and urine assay for diagnosis of opisthorchiasis with reference to primary or composite reference standards.**

| Locality | FECT | | | | Urine antigen assay | | | |
|---|---|---|---|---|---|---|---|---|
| | Primary standard[1] | | Composite standard[2] | | Primary standard[3] | | Composite standard[2] | |
| | Sens (%) (95% CI) | NPV (%) (95% CI) | Sens (%) (95% CI) | NPV (%) (95% CI) | Sens (%) (95% CI) | NPV (%) (95% CI) | Sens (%) (95% CI) | NPV (%) (95% CI) |
| KKN | 90.2 (0.804–0.956) | 96.1 (0.919–0.983) | 91.0 (0.818–0.960) | 96.2 (0.919–0.983) | 91.5 (0.818–0.965) | 96.7 (0.926–0.986) | 92.3 (0.834–0.968) | 96.7 (0.926–0.986) |
| KSN | 46.4 (0.388–0.542) | 80.0 (0.760–0.835) | 50.0 (0.425–0.574) | 80.0 (0.760–0.835) | 86.8 (0.777–0.927) | 96.8 (0.943–0.982) | 93.4 (0.884–0.963) | 96.8 (0.943–0.982) |
| Both sites | 59.5 (0.530–0.656) | 84.6 (0.815–0.873) | 62.3 (0.560–0.681) | 84.6 (0.815–0.873) | 88.8 (0.827–0.931) | 96.7 (0.948–0.980) | 93.1 (0.890–0.957) | 96.7 (0.948–0.980) |

[1]Primary standard = diagnosis by urine assay for three consecutive days as a gold standard

[2]Composite standard = diagnosis by FECT and urine antigen assay for three consecutive days as a gold standard

[3]Primary standard = diagnosis by FECT for three consecutive days as a gold standard

Sens = sensitivity, NPV = negative predictive value

KKN; Muang district, Khon Kaen Province KSN; Nong Kung Sri district, Kalasin Province

urine assay. The antigen levels in urine significantly correlated with fecal egg counts (EPG). A considerable number of 639 participants (*n* = 98, 15.3%) were fecal egg negative but yielded a positive urine assay. The urine ELISA had greater diagnostic sensitivity and, given the consistent level of antigen in urine, is more reliable than FECT for diagnosis of opisthorchiasis.

The overall prevalence according to the urine assay (30.2%) was higher than for FECT (20.2%). Based on FECT, the prevalence in KKN (28%) was higher than in KSN (16.6%). This difference was also seen in the initial FECT screening of a larger sample (*n* = 1,471). By contrast, the prevalence according to the urine assay in the two study localities was comparable (28.3% in KKN vs. 31.1% in KSN) and not associated with prevalence as estimated by FECT. One possible explanation is that urine assays can return a positive result from individuals with low or negative fecal egg counts [16, 17]. A previous study showed that the urine assay and fecal examination were equally sensitive in high-transmission areas, while the urine assay was more sensitive than fecal examination in low-transmission areas [17]. Clearly, more studies in different endemic settings may shed light on the factors that affect the performance of urine assay in diagnosis of opisthorchiasis.

Screening of fecal samples using FECT for three consecutive days resulted in an increased case detection of about 7.5%-10.0% compared with a single day examination. The inter-day

**Table 3. Diagnostic agreement (kappa value) for detection of *O. viverrini* infection between FECT and urine assay per day and cumulatively over three consecutive days.**

| Diagnostic method | Daily kappa value (95% CI) | Cumulative three days kappa value (95% CI) |
|---|---|---|
| **FECT** | | |
| Day 1 vs Day 2 | 0.422 (0.312–0.518) | 0.824 (0.765–0.879) |
| Day 2 vs Day 3 | 0.389 (0.290–0.503) | 0.862 (0.814–0.909) |
| Day 3 vs Day 1 | 0.350 (0.245–0.451) | 0.693 (0.622–0.761) |
| **Urine antigen assay** | | |
| Day 1 vs Day 2 | 0.970 (0.950–0.988) | 0.985 (0.972–0.997) |
| Day 2 vs Day 3 | 0.954 (0.930–0.976) | 0.985 (0.970–0.997) |
| Day 3 vs Day 1 | 0.930 (0.899–0.954) | 0.970 (0.951–0.988) |
| **FECT and urine antigen assay** | | |
| Day 1 | 0.437 (0.367–0.501) | 0.437 (0.367–0.501) |
| Day 2 | 0.351 (0.280–0.414) | 0.524 (0.458–0.591) |
| Day 3 | 0.355 (0.283–0.421) | 0.621 (0.599–0.678) |

variability of fecal examination might be due to inconsistency of egg excretion by the adult worms, particularly in the case of low worm burdens [10]. Certainly, over longer time spans (weeks and months), there can be considerable variation in numbers of eggs detected in feces [21]. Thus, diagnosis of opisthorchiasis by using multiple fecal examinations yields superior diagnostic performance compared to examination of a single stool sample [11]. However, this approach is laborious and is not practical for mass screening. In addition, it is not clear how many fecal examinations are required to assure a reliable diagnosis. In the current study, 7%-28% of individuals positive according to the urine assay were fecal egg negative. Under the scenario of widespread light infection, the reliability of a single fecal examination needs cautious interpretation [9, 16].

Unlike fecal examination, the urine assay had low inter-day variation in positive detection rates (0.8%-2.2%). This indicates that a single urine test is reliable for diagnosis of opisthorchiasis. Similar observations on daily variation in antigen concentration in urine have been examined for schistosomiasis [22–24]. Since collection of data for the urine assay in opisthorchiasis is in its initial phase, further studies in larger and more diverse localities with varying degrees of transmission are required.

We previously reported that antigen detection in urine was more sensitive than fecal examination for diagnosis of opisthorchiasis i.e.; 31–44% of positive antigen tests were in individuals with no observed fecal eggs [16, 17]. Similarly, in the current study 7–28% of fecal egg negative individuals had a positive urine assay. There are several supporting lines of evidence to suggest that these observations are not caused by false positive urine antigen results. Firstly, the results from a separate diagnostic, the copro-antigen test, have been shown to correlate with urine antigen results and detected cases in a similar proportion of fecal egg-negative individuals [17]. Secondly, praziquantel treatment in urine antigen positive individuals but fecal egg-negative resulted in a reduction of antigen levels in urine and these individuals became antigen-negative at 4 weeks post-treatment [18]. This observation suggests the role of adult worms, rather than eggs, as the source of antigen detected in urine in inidividuals with opisthorchiasis. A similar situation was previously observed in an autopsy study in which egg in feces were not detected in the majority of individuals with < 20 adult worms [10]. Lastly, an animal model study showed that the antigen detected in urine as well as in feces corresponded to the presence of worms in the liver of the *O. viverrini*-infected hamsters [25]. Moreover, after adult worms were killed by praziquantel treatment, the antigen test became negative in hamsters with complete worm clearance and the antigen test remained positive in hamsters with incomplete worm clearance. Therefore, based on the above evidence in human and experimental opisthorchiasis we argue that positive antigen detection in urine in fecal egg negative individuals represent active *O. viverrini* infection and we highlight that, in addition to qualitative diagnosis of opisthorchiasis, the urine antigen test can also be used quantitatively to assess the outcomes of drug treatment in individual patients or as part of a control program.

Confirmation of opisthorchiasis diagnosis by repeated fecal examination or worm expulsion is possible but impractical. Indeed, the expulsion chemotherapy may not be efficient since there were cases of fecal egg positive detections but no worm was recovered after treatment [19]. The autopsy study, however, showed an opposite scenario where fecal egg examinations were negative in the cases with worms (<20) in the liver [10]. In the present study, 3-daily fecal examination yielded 20.2% positive detection rates which was lower than 30.3% positive rate by urine antigen assay. In this regard, a new gold standard for diagnostic accuracy of opisthorchiasis deserves more investigation i.e. combining antigen and antibody detection with fecal examination.

The pathway of *O. viverrini* antigen excretion in urine is not completely understood. Antigen has been detected in urine and feces in infected individuals and has also been detected in

inflamed biliary tissue, where it became more pronounced after drug treatment in the hamster model [26]. Previous studies have also demonstrated a pathological change in kidneys and the presence of immune complexes in kidneys of *O. viverrini*-infected hamsters [27, 28]. Recently, the presence of *O. viverrini* antigen in kidney tissue has been implicated in chronic kidney disease [29]. Although further work on the pathway and circulation of *O. viverrini* antigen is required, it is likely that antigen produced by worms in the bile ducts enters blood circulation and filtered through the kidneys before being excreted in urine.

This study has provided further support for the reliability and usefulness of the urine antigen-detection assay for diagnosis of opisthorchiasis, particularly in areas with low transmission levels. The urine antigen assay can compensate for the limitations of fecal examination, since a single urine assay can identify light infections. In addition to adult worms, *O. viverrini* antigen in urine and feces may originate from juvenile worms as early as one week after infection in an opisthorchiasis-animal model [25]. Further investigation is needed to determine whether a single-worm infection can be detected using urine ELISA, as has been reported in schistosomiasis [30], since this is a critical threshold to assess the impact of parasite control and elimination programs.

There are several limitations to our study. Firstly, the extent of daily variation in antigen levels across longer periods (i.e. weeks or months) was not investigated and requires further study. Secondly, since the antigen concentration in feces is higher than in urine [15, 17], additional copro-antigen detection methods should be developed to validate the urine assay, particularly in egg-negative individuals who are positive by urine assays. Thirdly, the nature of *O. viverrini* antigen detected in urine is currently not known. Whether the antigen in urine is in the form of immune complexes or is free antigen deserves further study. Lastly, the main drawback of urine assay by ELISA is that it takes time to complete and needs experienced operators and special equipment. Adaptation of the urine assay for rapid-test platforms [31, 32], including rapid diagnostic tests, is needed for large scale screening of opisthorchiasis.

## Conclusion

Community-based diagnosis of opisthorchiasis in Northeast Thailand revealed that antigen detection by the urine assay was more sensitive than fecal examination for parasite eggs using FECT. Consistently high diagnostic performances were observed in localities with differing prevalence of *O. viverrini* infection. The inter-day variation in the estimated prevalence of opisthorchiasis, as determined using the urine assay, was low (2%) while a significantly greater daily variation (7.5%) was seen in results from FECT. A significant correlation was observed between *O. viverrini* antigen concentration in urine and fecal egg counts. As opposed to fecal examination, the urine assay had higher diagnostic sensitivity and consistency, which suggests that a single urine assay is reliable for diagnosis of opisthorchiasis. Collection of urine is considered less invasive by the target population, further supporting the use of the urine antigen assay as an effective tool for diagnosis and screening of opisthorchiasis. Further development of the urine assay into a rapid point-of-care test platform is currently being undertaken to facilitate the large-scale screening of opisthorchiasis to achieve control in the Mekong region.

## Supporting information

**S1 File. Raw data for Fig 1.**
(XLSX)

**S2 File. Raw data for Fig 2.**
(XLSX)

**S3 File. Raw data for Fig 3.**
(XLSX)

**S1 Table. Baseline prevalence of parasitic infection among project participants (*n* = 1,471) as determined by FECT and stratified by locality (Muang District, Khon Kaen Province (KKN, *n* = 448) and Nong Kung Sri District, Kalasin Province (KSN, *n* = 1,023).**
(DOCX)

**S2 Table. The age- and sex-prevalence of *O. viverrini* infection determined by FECT and urine assay in Muang District, Khon Kaen Province (KKN), Nong Kung Sri District, Kalasin Province (KSN) (*n* = 1,471).**
(DOCX)

**S3 Table. Relationships between antigen concentrations of *O. viverrini* and intensity of infection (EPG) determined by FECT.** Analyses using linear regression models were performed based on each locality (KKN and KSN) separately and combined.
(DOCX)

**S4 Table. Positive rates of *O. viverrini* infection determined by urine antigen detection assay in groups of participants stratified by intensity of infection (EPG).** KKN: Muang District, Khon Kaen Province, KSN: Nong Kung Sri District, Kalasin Province.
(DOCX)

## Acknowledgments

We would like to acknowledge Prof. David Blair for editing the manuscript via Publication Clinic KKU, Thailand.

## Author Contributions

**Conceptualization:** Chanika Worasith, Chatanun Eamudomkarn, Paiboon Sithithaworn.

**Data curation:** Chanika Worasith, Phattharaphon Wongphutorn, Chutima Homwong.

**Formal analysis:** Chanika Worasith, Phattharaphon Wongphutorn, Chutima Homwong.

**Funding acquisition:** Narong Khuntikeo, Watcharin Loilome, Thomas Crellen, Paiboon Sithithaworn.

**Investigation:** Chanika Worasith, Phattharaphon Wongphutorn, Chutima Homwong, Kulthida Y. Kopolrat.

**Methodology:** Chanika Worasith, Anchalee Techasen, Raynoo Thanan, Chatanun Eamudomkarn, Chompunoot Wangboon, Jiraporn Sithithaworn.

**Project administration:** Kulthida Y. Kopolrat, Paiboon Sithithaworn.

**Resources:** Anchalee Techasen, Raynoo Thanan, Chatanun Eamudomkarn, Chompunoot Wangboon, Watcharin Loilome.

**Software:** Anchalee Techasen, Raynoo Thanan, Chatanun Eamudomkarn, Chompunoot Wangboon.

**Supervision:** Narong Khuntikeo, Jiraporn Sithithaworn, Paiboon Sithithaworn.

**Validation:** Chanika Worasith, Jiraporn Sithithaworn, Thomas Crellen, Paiboon Sithithaworn.

**Visualization:** Chanika Worasith, Phattharaphon Wongphutorn, Chutima Homwong, Kulthida Y. Kopolrat.

**Writing – original draft:** Chanika Worasith.

**Writing – review & editing:** Thomas Crellen, Paiboon Sithithaworn.

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
