## [Decision Letter · Decision Letter 0]

17 Jun 2022

PONE-D-22-13305Effects of day-to-day variation of Opisthorchis viverrini antigen in urine on the accuracy of diagnosis of opisthorchiasis in Northeast ThailandPLOS ONE

Dear Dr. Sithithaworn,

Thank you for submitting your manuscript to PLOS ONE. After careful consideration, we feel that it has merit but does not fully meet PLOS ONE’s publication criteria as it currently stands. Therefore, we invite you to submit a revised version of the manuscript that addresses the points raised during the review process.

Address the few suggestions raised by Reviewer 1.

We look forward to receiving your revised manuscript.

Kind regards,

Marcello Otake Sato, Ph.D., D.V.M.

Academic Editor

PLOS ONE

Journal Requirements:

Additional Editor Comments:

The MS is well written and addresses an important point on opisthorchiasis, that is the improvement of diagnosis tests. Hope the authors can address the few comments by the Reviewer 1, improving the Discussion.

Reviewers' comments:

Reviewer's Responses to Questions

**Comments to the Author**

1. Is the manuscript technically sound, and do the data support the conclusions?

Reviewer #1: Yes

Reviewer #2: Yes

2. Has the statistical analysis been performed appropriately and rigorously? 

Reviewer #1: Yes

Reviewer #2: Yes

3. Have the authors made all data underlying the findings in their manuscript fully available?

Reviewer #1: Yes

Reviewer #2: Yes

4. Is the manuscript presented in an intelligible fashion and written in standard English?

Reviewer #1: Yes

Reviewer #2: Yes

5. Review Comments to the Author

Reviewer #1: The validity of an alternative urine ELISA on day-to-day change of opisthorchiasis compared to FECT was described in this paper. This is an important issue that affects the diagnosis of opisthorchiasis and other helminthic infections. Because the manuscript was well-organized and discussed, it yielded significant results. There are only a few comments that need to be addressed.

Discussion

1) L327: “The urine ELISA had greater diagnostic sensitivity and, given the consistent level of antigen in urine, is more reliable than FECT for diagnosis of opisthorchiasis.”

How to ensure that the greater sensitivity of urine ELISA in this study did not come from the false positive. To be confirm, FECT more than three days may need to be consecutively performed or maybe worm expulsion examination or any alternative method for definite diagnosis. Please discuss more this point.

2) Are there any evidence on how long urine ELISA stay positive after treatment? Did the authors collect the data on previous anthelminthic treatment of the subjects?

Reviewer #2: Comments to the manuscript PONE-D-22-13305, entitled "Effects of day-to-day variation of Opisthorchis viverrini antigen in urine on the accuracy of diagnosis of opisthorchiasis in Northeast Thailand" by Chanika Worasith et al.

My pleasure to read such good manuscript: evidences expertise and domain of concepts, originality and a lot of work in order to a better well-being of Thai endemic population for O. viverrini. In my point of view this manuscript is in good shape for publication. In addition, it seems to me to accomplish PLOS ONE, criteria: original research; original results; Experiments, statistics, and other analyses are described in sufficient detail; conclusions are presented in an appropriate fashion and are supported by the data; the article is presented in an intelligible fashion and is written in standard English (I am not a English native); the research meets all applicable standards for the ethics of experimentation and research integrity; finally, the article adheres to appropriate reporting guidelines and community standards for data availability.

6. PLOS authors have the option to publish the peer review history of their article (what does this mean?). If published, this will include your full peer review and any attached files.

Reviewer #1: No

Reviewer #2: **Yes: **José M. Correia da Costa

---

## [Author Response · Author response to Decision Letter 0]

30 Jun 2022

Title: Effects of day-to-day variation of Opisthorchis viverrini antigen in urine on the accuracy of diagnosing opisthorchiasis in Northeast Thailand

(PONE-D-22-13305)

Response to Editor (green label)

1. Please ensure that your manuscript meets PLOS ONE's style requirements. 

Answer: Yes, the manuscript was constructed using the PLOS ONE style templates.

Answer: The text has been modified as “Written informed consent was obtained from each project participant” (page 5, line 155-156). 

Answer: Yes, we have checked the reference list and all references are complied with PLOS ONE’s requirements. There is no cited reference that has been retracted in the reference list.

Response to reviewers

Reviewer: 1 (yellow label)

Discussion

1) L327: “The urine ELISA had greater diagnostic sensitivity and, given the consistent level of antigen in urine, is more reliable than FECT for diagnosis of opisthorchiasis.”

How to ensure that the greater sensitivity of urine ELISA in this study did not come from the false positive. To be confirm, FECT more than three days may need to be consecutively performed or maybe worm expulsion examination or any alternative method for definite diagnosis. Please discuss more this point.

Answer: We have added the point raised regarding the greater sensitivity of urine ELISA in this study and the confirmation of antigen detection by definite diagnostic methods in Discussion section line 362-396.

We previously reported that antigen detection in urine was more sensitive than fecal examination for diagnosis of opisthorchiasis i.e.; 31-44% of positive antigen tests were in individuals with no observed fecal eggs [16, 17]. Similarly, in the current study 7-28% of fecal egg negative individuals had a positive urine assay. There are several supporting lines of evidence to suggest that these observations are not caused by false positive urine antigen results. Firstly, the results from a separate diagnostic, the copro- antigen test, have been shown to correlate with urine antigen results and detected cases in a similar proportion of fecal egg-negative individuals [17]. Secondly, praziquantel treatment in urine antigen positive individuals but fecal egg-negative resulted in a reduction of antigen levels in urine and these individuals became antigen-negative at 4 weeks post-treatment [18]. This observation suggests the role of adult worms, rather than eggs, as the source of antigen detected in urine in inidividuals with opisthorchiasis. A similar situation was previously observed in an autopsy study in which egg in feces were not detected in the majority of individuals with < 20 adult worms [10]. Lastly, an animal model study showed that the antigen detected in urine as well as in feces corresponded to the presence of worms in the liver of the O. viverrini-infected hamsters [25]. Moreover, after adult worms were killed by praziquantel treatment, the antigen test became negative in hamsters with complete worm clearance and the antigen test remained positive in hamsters with incomplete worm clearance. Therefore, based on the above evidence in human and experimental opisthorchiasis we argue that positive antigen detection in urine in fecal egg negative individuals represent active O. viverrini infection and we highlight that, in addition to qualitative diagnosis of opisthorchiasis, the urine antigen test can also be used quantitatively to assess the outcomes of drug treatment in individual patients or as part of a control program.

Confirmation of opisthorchiasis diagnosis by repeated fecal examination or worm expulsion is possible but impractical. Indeed, the expulsion chemotherapy may not be efficient since there were cases of fecal egg positive detections but no worm was recovered after treatment [19]. The autopsy study, however, showed an opposite scenario where fecal egg examinations were negative in the cases with worms (<20) in the liver [10]. In the present study, 3-daily fecal examination yielded 20.2% positive detection rates which was lower than 30.3% positive rate by urine antigen assay. In this regard, a new gold standard for diagnostic accuracy of opisthorchiasis deserves more investigation i.e. combining antigen and antibody detection with fecal examination. 

2) Are there any evidence on how long urine ELISA stay positive after treatment? Did the authors collect the data on previous anthelminthic treatment of the subjects?

Answer: In our previous study (Worasith et al., 2020), we applied the urine antigen assay to monitor the antigen profile after praziquantel treatment in individuals with opisthorchiassis. The percentage of individuals that were urine antigen positive declined to 17.7% at two weeks and were all negative (0%) four weeks after treatment. Data in this study also showed that in the antigen positive group (n=12), the antigen positivity rate and the antigen concentration were not change when determined at an interval of 11 months. In the current study, the antigen levels in urine showed no significant change over 3 days period. These data suggest the stability and consistency of antigen levels in urine in active opisthorchiasis. See answer #1.

We collected data on individuals with past anthelminthic treatment and found no correlation with the results of urine antigen assay (unpublished data).

Reviewer: 2

Comments to the manuscript PONE-D-22-13305, entitled "Effects of day-to-day variation of Opisthorchis viverrini antigen in urine on the accuracy of diagnosis of opisthorchiasis in Northeast Thailand" by Chanika Worasith et al. My pleasure to read such good manuscript: evidences expertise and domain of concepts, originality and a lot of work in order to a better well-being of Thai endemic population for O. viverrini. In my point of view this manuscript is in good shape for publication. In addition, it seems to me to accomplish PLOS ONE, criteria: original research; original results; Experiments, statistics, and other analyses are described in sufficient detail; conclusions are presented in an appropriate fashion and are supported by the data; the article is presented in an intelligible fashion and is written in standard English (I am not a English native); the research meets all applicable standards for the ethics of experimentation and research integrity; finally, the article adheres to appropriate reporting guidelines and community standards for data availability.

Answer: We thank the reviewer for their kind remarks. 

Author’s corrections: (blue label) for the tittle and in the text.

---

## [Editor Report · Decision Letter 1]

4 Jul 2022

Effects of day-to-day variation of Opisthorchis viverrini antigen in urine on the accuracy of diagnosing opisthorchiasis in Northeast Thailand

PONE-D-22-13305R1

Dear Dr. Sithithaworn,

We’re pleased to inform you that your manuscript has been judged scientifically suitable for publication and will be formally accepted for publication once it meets all outstanding technical requirements.

Kind regards,

Marcello Otake Sato, Ph.D., D.V.M.

Academic Editor

PLOS ONE
---

## [Editor Report · Acceptance letter]

10 Jul 2022

PONE-D-22-13305R1 

Effects of day-to-day variation of *Opisthorchis viverrini* antigen in urine on the accuracy of diagnosing opisthorchiasis in Northeast Thailand 

Dear Dr. Sithithaworn:

I'm pleased to inform you that your manuscript has been deemed suitable for publication in PLOS ONE. Congratulations! Your manuscript is now with our production department. 

Kind regards, 

on behalf of

Dr. Marcello Otake Sato 

Academic Editor

PLOS ONE